

# Utilization of hepatitis B surface antigen-positive donors in liver transplantation for recipients with hepatocellular carcinoma: a retrospective and propensity score matching analysis

Zhitao Chen[*], Yihao Ma[*], Yuqi Dong, Chuanbao Chen, Hanyu Wang, Tielong Wang, Jia Yu, Xitao Hong, Maogen Chen, Xiaoshun He and Weiqiang Ju

[1] Organ Transplant Center, First Affiliated Hospital of Sun Yat-sen University, Guangzhou, Grangdong, People's Republic of China
[2] Guangdong Provincial Key Laboratory of Organ Donation and Transplant Immunology, Guangzhou, Guangdong, People's Republic of China
[*] These authors contributed equally to this work.

Corresponding authors
Xiaoshun He, gdtrc@163.com
Weiqiang Ju, weiqiangju@163.com

## ABSTRACT

**Introduction**. The use of extended criteria donor (ECD) grafts such as donor with infection of hepatitis B virus (HBV) is a potential solution for organ shortage. In this study, we aimed to evaluate the safety and long-term survival of utilization of hepatitis B surface antigen-positive (HBsAg+) donor livers in HCC patients using propensity score matching (PSM) analysis.

**Methods**. Forty-eight donors with HBsAg-positive and 279 donors with HBsAg-negative were transplanted and enrolled in this study. PSM analysis were used to eliminate selection bias. Perioperative data and survival were collected and analyzed.

**Results**. PSM generated 44 patient pairs. When comparing intra- and post-operative data, no significant difference was found between groups ($P > 0.05$). Patients with a HBsAg-positive donor had significantly worse progression-free survival (1-year: 65.9% vs. 90.9%; 3-year: 18.1% vs. 70.4%, $P = 0.0060$) and overall survival (1-year: 84.1% and 95.4%; 3-year: 27.2% vs. 79.5%, $P = 0.0039$). In multivariate analysis, donor HBsAg-positivity was an independent risk factor for survival and occurrence ($P = 0.005$ and $0.025$, respectively).

**Conclusion**. In conclusion, with adequate antiviral prophylaxis and treatment, utilization of HBsAg positive liver grafts did not increase the incidence of early-stage complications. However, patient with an HBsAg-positive graft had poorer progression-free survival and overall survival.

## INTRODUCTION

At present, liver transplantation (LT) has been regarded as the only therapeutic option for patients with end-stage liver diseases. In China, the number of patients on liver transplant waiting lists is increasing, and the organ shortage crisis is obvious (*Hou et al., 2015*). There are not enough donor organs available to meet the needs of the growing number of patients on the waiting list resulting in thousands of deaths every year (*Goldaracena et al., 2020*). One possible solution to expand the donor pool is to take extended criteria donor (ECD) grafts such as donor seropositivity for hepatitis B virus (HBV) for consideration (*Hashimoto & Miller, 2008*).

HBV infection is a worldwide health-threatening problem. Globally, 257 million people are living with HBV infection (*Schweitzer et al., 2015*). In China, there are 100 million HBV carriers (*Gao et al., 2019*). Donor seropositivity for HBV has long been considered a contraindication to LT, for that may cause HBV infection or transmission in recipients (*Hashimoto & Miller, 2008*). Nowadays, with the application of hepatitis B immune globulin (HBIG) and anti-viral treatments, the incidence of graft reinfection with HBV is drastically reduced (*Park, Gayam & Pan, 2020*). The utilization of donor seropositivity for HBV can greatly alleviate the severe organ shortage and benefit more patients on the transplant waiting list. Indeed, some previous studies have demonstrated that deceased donor livers positive for hepatitis B surface antigen (HBsAg) can be used in HBsAg positive or negative recipients. In a multicenter study in Italy, 28 patients with HBsAg positive grafts were followed for a mean post-transplant survival of 63.7 months with HBIG and antiviral treatment, and no primary non-function (PNF), re-LT, early or late complications including hepatic artery thrombosis (HAT) occurred after LT (*Ballarin et al., 2017*). Data from a retrospective study in Asia comparing 42 patients undergoing LT with HBsAg positive donor grafts and 327 patients with HBsAg negative ones, suggested similar outcomes in terms of graft function recovery, complications and comparable graft survivals (*Yu et al., 2014*). Similar conclusions were derived from data reported by other authors in another multicentric retrospective study in Asia, which showed that the 1-, 3- and 5-year HBV recurrence of patients with HBsAg-positive donor was higher compared with the HBsAg negative ones (*Wei et al., 2018*).

Hepatocellular carcinoma (HCC) is one of the main indications to LT. The primary etiology associated with HCC development is HBV, particularly in developing countries (*Li et al., 2019*). Currently, reports on the utilization of HBsAg-positive liver grafts in patients with HCC are quite scanty.

To further clarify the current situation, we aimed to evaluate the safety and long-term efficacy of the utilization of HBsAg+ liver grafts in patients with HCC using propensity score matching (PSM) analysis.

## MATERIAL AND METHODS

This was a retrospective analysis from a single center. The study was approved by the Ethics Committee of the First Affiliated Hospital of Sun Yat-sen University (Ethics Committee Approval Document Number: (2021)437). All procedures conformed to the
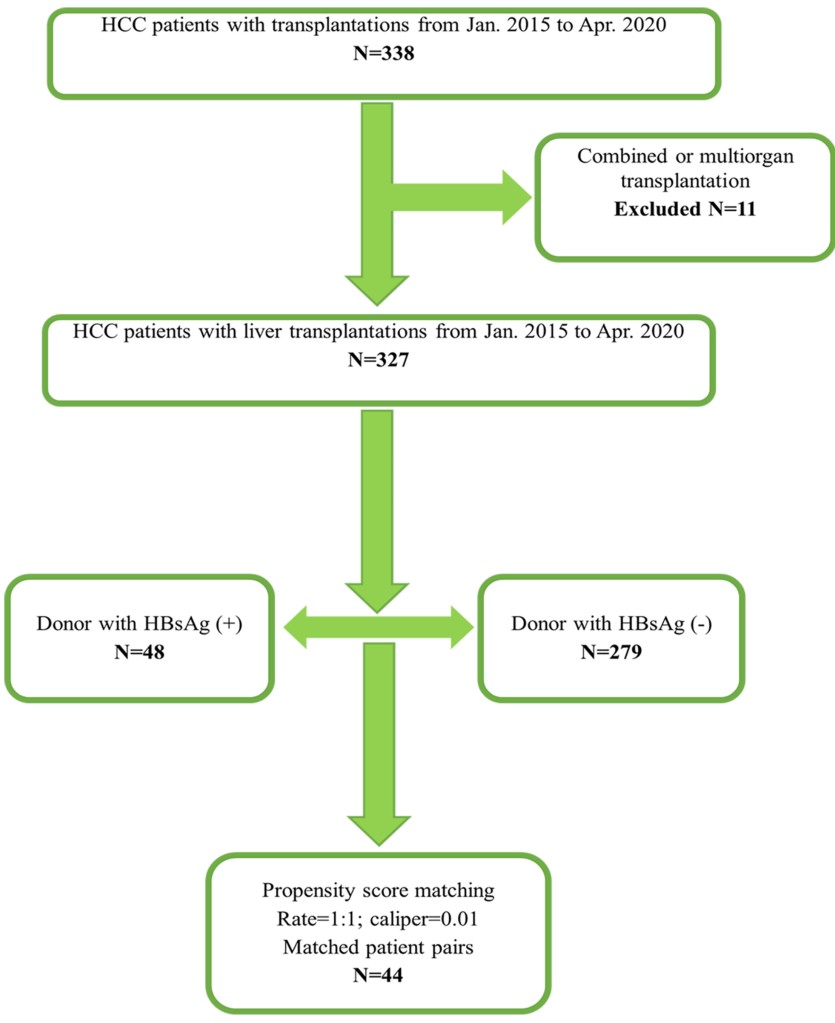

**Figure 1  Flowchart of this study.**

ethical guidelines of the Declaration of Helsinki (1975) and informed consent waiver was granted by the IEC because of the retrospective, minimal risk nature of the study. All organs were from donation and no organs from executed prisoners were used for transplantation in this study.

## Patients enrollment

From January 2015 to April 2020, 338 patients were diagnosed with HCC and underwent deceased organ transplantations in our center. After excluded 11 combined or multi-organ transplantations, a total of 327 patients diagnosed with HCC and then underwent LT were retrospectively reviewed and enrolled in this study. Forty-eight HBsAg-positive donors and 279 HBsAg-negative donors were transplanted. The flow chart of this study is shown in Fig. 1.

## PSM analysis

Because HBsAg-positive donor was not randomly assigned to patients with HCC, PSM was used to minimize patient selection bias and eliminate potential confounders (*Badhiwala, Karmur & Wilson, 2021*). Patients in the donor HBsAg-positive group were paired with those with HBsAg-negative donors at 1:1 ratio. Nearest-neighbor matching algorithm was adopted (caliper width = 0.01). Variables that may influence the clinical posttransplant outcome were included in the PSM model, which included the following variables (*Pandyarajan, Govalan & Yang, 2021*): donor gender (male or female) and age, donor type (DBD or DCD), body mass index (BMI), donor's Aspartate aminotransferase (AST), Alanine aminotransferase (ALT) and bilirubin, recipient gender (male or female) and age, pretransplant HBV-DNA (>100 or ≤ 100 copies /ml),pretransplant HBsAg(positive or negative), recipient BMI, Model for end-stage liver disease (MELD) score, Child-Pugh score, alpha feto-protein (AFP) level (≤400 or >400 ng/mL) , pretransplant treatment, TNM staging (I or II to III).

## Post-transplant therapy and follow-up monitoring

The immunosuppressive regimen all patients received after LT was tacrolimus +mycophenolate mofetil. Basiliximab was used for immune induction with a dose of 20 mg by intravenous injection before abdominal closure and on posttransplant day 4. The target drug concentration of tacrolimus was 8 to 12 ng/ml in the first post-operative month and aimed at average of 5–7 ng/ml thereafter.

Antiviral prophylaxis and hepatitis B immunoglobulin therapy were given to the patients undergoing LT. Briefly, 5000 IU HBIG was given intravenously at the time of transplant during the anhepatic phase, followed by 2500 IU intravenous injection once in the first week and then weekly for the next 3 weeks, and 400 IU intramuscularly monthly. Antiviral treatment was with entecavir orally administrated daily.

Postoperative visits will be performed on postoperative day (POD) 1–7, POD 14 and each month post-transplantation. Biomedical values, complications, adverse events and medication administration records will be documented. Ultrasound doppler examination of vascular graft patency was performed was performed once every 2 days for 7 days. Serovirological test including HBsAg and HBVDNA was monitored at day 7, 14, 30 and then monthly. HBsAg more than 0.05 IU/ml was detectable and considered positive. HBV- DNA more than $1.0 \times 10^2$ copies/ml was detectable and considered positive. Diagnosis of tumor recurrence was based on imaging studies, clinical findings and laboratory tests and diagnosed according to Guidelines for the Diagnosis and Treatment of Primary Liver Cancer (2019 edition) in China (*Qiu et al., 2020*). The date of last follow-up or death was used for follow-up and Kaplan–Meier analysis, and the time to recurrence was calculated from the date of surgery to the date of recurrence or last follow-up.

## Statistical analysis

All the statistical analyses of the data were performed using SPSS Version 26.0 (SPSS Inc., Chicago, IL, USA) and GraphPad Prism Version 9.0.1 (GrapPad, San Diego, CA,

**Table 1  Baseline characteristics in HCC patients.**

| Variables | Before propensity matching | | | After propensity matching | | |
|---|---|---|---|---|---|---|
| | HBsAg-positive (N = 48) | HBsAg-negative (N = 279) | P | HBsAg-positive (N = 44) | HBsAg-negative (N = 44) | P |
| **Donors' characteristics** | | | | | | |
| Sex, Male n (%) | 42 (87.5) | 210 (75.3) | 0.063 | 38 (86.4) | 37 (84.1) | 0.764 |
| Age (ys) | 37.08 ± 1.58 | 37.09 ± 0.90 | 0.996 | 37.05 ± 1.69 | 34.34 ± 2.16 | 0.133 |
| Donor type, DBD, n (%) | 34 (70.8) | 217 (77.8) | 0.293 | 31 (70.5) | 32 (72.7) | 0.813 |
| BMI (kg/m2) | 22.63 ± 0.38 | 22.20 ± 0.18 | 0.364 | 22.48 ± 0.39 | 21.35 ± 0.48 | 0.071 |
| AST(U/L) | 121.02 ± 37.20 | 112.93 ± 9.21 | 0.765 | 70.54 ± 7.32 | 109.21 ± 20.53 | 0.080 |
| ALT(U/L) | 70.20 ± 14.07 | 78.70 ± 7.84 | 0.668 | 55.34 ± 7.10 | 55.05 ± 7.08 | 0.977 |
| Bilirubin (umol/L) | 21.48 ± 1.88 | 26.40 ± 1.54 | 0.045* | 21.67 ± 1.98 | 24.07 ± 2.84 | 0.490 |
| **Recipients' characteristics** | | | | | | |
| Recipient age, years | 50.58 ± 1.58 | 50.29 ± 0.67 | 0.866 | 50.00 ± 1.68 | 51.61 ± 1.57 | 0.484 |
| Recipient sex Male, n (%) | 45 (93.8) | 257 (92.1) | 0.694 | 41 (93.2) | 40 (90.9) | 0.694 |
| HBV-DNA (+), n (%) | 11 (22.9) | 52 (18.6) | 0.488 | 11 (25.0) | 8 (18.2) | 0.437 |
| HBsAg (+), n (%) | 48 (100.0) | 279 (100.0) | >0.999 | 44 (100.0) | 44 (100.0) | >0.999 |
| BMI (kg/m2) | 23.72 ± 0.48 | 23.20 ± 0.20 | 0.305 | 23.89 ± 0.50 | 23.91 ± 0.52 | 0.980 |
| MELD | 14.94 ± 1.57 | 14.48 ± 0.53 | 0.785 | 14.09 ± 1.57 | 13.05 ± 1.16 | 0.594 |
| Child-Pugh score | 7.15 ± 0.28 | 7.48 ± 0.12 | 0.412 | 7.09 ± 0.29 | 7.09 ± 0.28 | >0.999 |
| AFP >400, n (%) | 17 (35.4) | 91 (32.6) | 0.703 | 15 (34.1) | 8 (18.2) | 0.089 |
| RFA, n (%) | 9 (18.8) | 55 (19.7) | 0.877 | 8 (18.2) | 10 (22.7) | 0.597 |
| TACE, n (%) | 12 (25.0) | 86 (30.8) | 0.416 | 11 (25.0) | 12 (27.3) | 0.808 |
| Tumor grade II–III, n (%) | 30 (62.5) | 177 (63.4) | 0.901 | 26 (59.1) | 29 (65.9) | 0.509 |

Notes.

HBsAg, hepatitis B surface antigen; DBD, donor after brain death; BMI, body mass index; AST, Aspartate aminotransferase; ALT, aminotransferase; MELD, model for end-stage liver disease; AFP, alpha feto-protein; RFA, radiofrequency ablation; TACE, transarterial chemoembolization.

USA). Data are expressed as the mean ± standard deviation, median (range) or number (percentage). Categorical variables including gender, donor type, HBsAg positive or negative, HB-DNA positive or negative, tumor TNM staging were analyzed by chi-square tests, and continuous variables including age, value of AST, ALT and bilirubin, MELD score and Child-pugh score were analyzed by Fisher's exact test. In addition, a Cox proportional hazards model was constructed for multivariate analysis. For survival analysis, OS and PFS were calculated by the Kaplan–Meier method with a log-rank test. A $P$ value of less than 0.05 was considered significant.

# RESULTS

## Study population and patient characteristics

The baseline data of the two groups were shown in Table 1. Before PSM, significant differences of the recipients were noted in donor pretransplant bilirubin (21.48 ± 1.88 *vs.* 26.40 ± 1.54 umol/L, $P = 0.045$). Using PSM, we generated 44 matched pairs of patients. No significant difference was found between the HBsAg positive and negative donor groups in the comparison of all clinical characteristics ($P > 0.05$).

**Table 2  Intraoperative and postoperative data of patients with HCC after PSM.**

|  | HBsAg-positive (N = 44) | HBsAg-negative (N = 44) | P |
|---|---|---|---|
| Anhepatic time (mins) | 55.43 ± 2.41 | 58.71 ± 2.47 | 0.345 |
| Cold ischemia time (hrs) | 7.02 ± 0.44 | 7.41 ± 0.41 | 0.520 |
| Operation duration (hrs) | 7.45 ± 0.26 | 7.98 ± 0.23 | 0.129 |
| RBC volume (IU) | 5.54 ± 0.86 | 5.56 ± 0.93 | 0.989 |
| FFP volume (IU) | 6.99 ± 0.67 | 7.56 ± 0.67 | 0.555 |
| Platelet volume (u) | 0.54 ± 0.28 | 0.51 ± 0.10 | 0.939 |
| Blood loss (ml) | 2275.40 ± 405.64 | 1859.77 ± 233.39 | 0.377 |
| Cardiac arrest, n (%) | 0 (0) | 1 (2.3) | 0.315 |
| Intraoperative hypotension, n (%) | 3 (6.8) | 2 (4.5) | 0.645 |
| Ventilation time (hrs) | 47.79 ± 13.18 | 35.36 ± 11.61 | 0.481 |
| ICU stay time (hrs) | 64.61 ± 12.09 | 52.97 ± 13.77 | 0.492 |
| Abdominal bleeding, n (%) | 1 (2.3) | 0 (0) | 0.315 |
| HAT, n (%) | 0 (0) | 4 (9.0) | 0.153 |
| PVT, n (%) | 0 (0) | 1 (2.3) | 0.315 |
| EAD, n (%) | 16 (36.4) | 21 (47.7) | 0.280 |
| PNF, n (%) | 2 (4.5) | 0 (0) | 0.153 |
| Death in 30 days, n (%) | 2 (4.5) | 0 (0) | 0.153 |
| HBsAg (+), n (%) | 3 (6.8) | 0 (0) | 0.078 |
| HBV DNA (+), n (%) | 1 (2.3) | 0 (0) | 0.315 |
| Tumor recurrence n (%) | 18 (40.9) | 12 (27.3) | 0.177 |

**Notes.**

HBsAg, hepatitis B surface antigen; RBC, red blood cell; FFP, fresh frozen plasma; ICU, intensive care unit; HAT, hepatic artery thrombosis; PVT, portal vein thrombosis; EAD, early allograft dysfunction; PNF, primary nonfunction.

## Post-PSM assessment of the perioperative outcomes of HBsAg-positive donors

The intratransplant and posttransplant data of two groups are presented in Table 2. No significant differences were found in anhepatic time($P = 0.345$), cold ischemia time ($P = 0.520$) and operation duration( $P = 0.129$; Fig. 2A). In comparison of intratransplant transfusion, no significant differences were found in blood loss ($P = 0.377$), RBC transfusion ($P = 0.989$), FFP transfusion ($P = 0.555$) and platelet transfusion ($P = 0.939$), either (Fig. 2B). One patient suffered cardiac arrest and 5 patients suffered hypotension during operation ($P = 0.315$ and 0.645, respectively). After transplantation, the duration of mechanical ventilation and ICU stay were not significantly different between the two groups ($P = 0.481$ and 0.492, respectively).In addition, we observed that there was no difference when comparing the major early-stage complications between the two groups. No statistical significance was found in the incidence of abdominal bleeding (1 *vs.* 0, $P = 0.315$), HAT (0 *vs.* 4, $P = 0.153$), portal vein thrombosis (PVT, 0 *vs.* 1, $P = 0.315$), early allograft dysfunction (EAD, 16 *vs.* 21 $P = 0.280$) and primary nonfunction (2 *vs.* 0, $P = 0.153$). The 30-day mortality rate was 4.5% and 0 in the HBsAg-positive and HBsAg-negative allograft groups, respectively ($P = 0.153$). The result of HBV recurrence showed that HBsAg was detectable in three patients and HBV DNA was detectable in 1 patient in

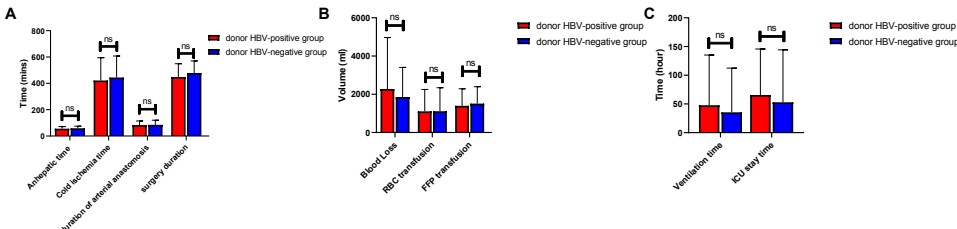

**Figure 2** **Comparison of intraoperative and postoperative data between groups.** (A) Anhepatic time, cold ischemia time, duration of arterial anastomosis and surgery duration. (B) Blood loss, RBC transfusion and FFP transfusion. (C) Ventilation time and ICU stay time.

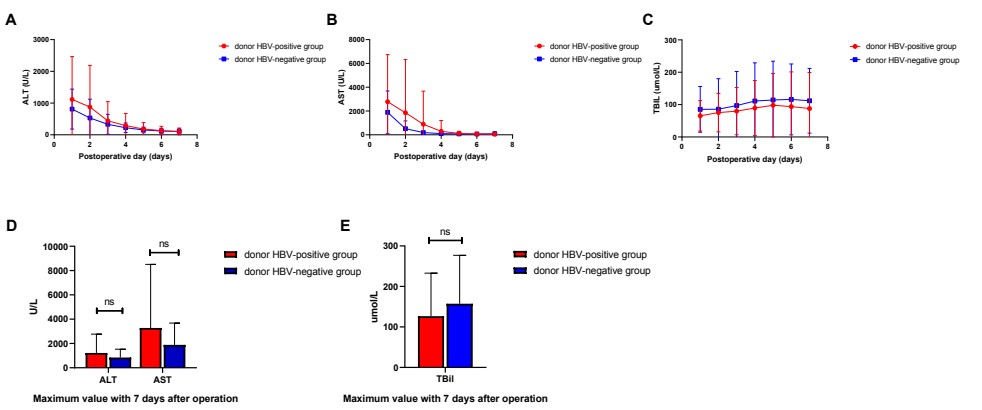

**Figure 3** **Postoperative liver function recovery between groups.** (A) Changes of Alanine aminotransferase (ALT) within 7 days; (B) changes of Aspartate aminotransferase (AST) within 7 days; (C) changes of total bilirubin with 7 days. (D) Maximum value of ALT and AST within 7 days. (E) Maximum value of total bilirubin within 7 days.

HBsAg-positive graft groups, respectively, without significant differences ($P = 0.078$ and 0.315, respectively).

Dynamic changes of postoperative liver function recovery between groups are presented in Fig. 3. Posttransplant monitoring results showed that ALT and AST dropped to normal level in day 4 after transplantation, however total bilirubin level remained high in 7 days after transplantation (Figs. 3A–3C). Maximum values of ALT, AST and TBIL within 7 days were not significantly higher in patients with HBsAg-positive liver grafts compared with those in HBsAg-negative liver grafts group ($1224.75 \pm 232.88$ *vs.* $841.77 \pm 103.48$ IU/L, $P = 0.138$; $3281.00 \pm 788.66$ *vs.* $1888.48 \pm 270.49$ IU/L, $P = 0.101$; $126.60 \pm 16.04$ *vs.* $157.377 \pm 18.00$ umol/L, $P = 0.205$; Fig. 3E).

## Survival and multivariant analysis between groups

During follow-up, tumor recurrence occurred in 18 of 44 (40.9%) and 12 of 44 (27.2%) patients in the donor HBsAg-positive and HBsAg-negative groups, respectively (Table 2). Based on Kaplan–Meier survival analysis, the 1-year and 3-year PFS rates were 65.9% and 18.1% in the donor HBsAg-positive group, respectively, compared with 90.9% and 70.4%

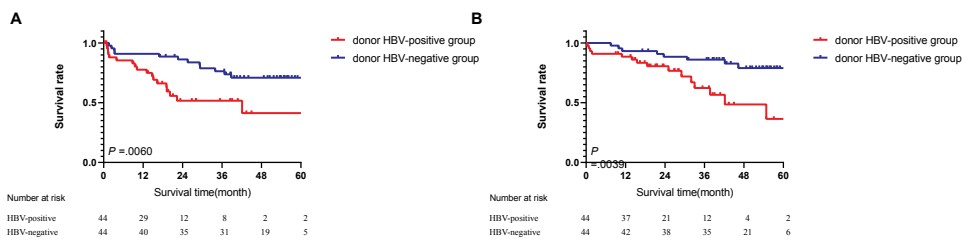

**Figure 4** **Survival analysis between groups.** (A) Progression-free survival. (B) Overall survival.

**Table 3** **Multivariant analyses for independent risk factors of recurrence and overall survival.**

| Variants | Multivariant analysis | | | |
|---|---|---|---|---|
| | Survival | | Occurrence | |
| | RR (95%CI) | *P* | RR (95%CI) | *P* |
| Donor HBsAg-positive | 2.306 (1.279~4.155) | 0.005[*] | 2.429 (1.115~5.289) | 0.025[*] |
| MELD >15 | 1.156 (0.489~2.734) | 0.742 | 2.348 (0.756~7.292) | 0.140 |
| Child-Pugh score >7 | 0.421 (0.213~0.833). | 0.013[*] | 0.981 (0.345~2.794) | 0.972 |
| AFP >400 | 0.996 (0.511~1.939) | 0.990 | 1.962 (0.878~4.383) | 0.100 |
| Tumor grade II–III | 1.345 (0.749~2.414) | 0.321 | 1.735 (0.709~4.246) | 0.227 |
| EAD | 1.013 (0.568~1.805) | 0.965 | 0.786 (0.344~4.798) | 0.569 |
| Tumor recurrence | 0.617 (0.314~1.211) | 0.160 | | |

**Notes.**
HBsAg, hepatitis B surface antigen; MELD, model for end-stage liver disease; AFP, alpha feto-protein; EAD, early allograft dysfunction.

in the donor HBsAg-negative group, respectively, and the differences were statistically significant ($P = 0.0060$; Fig. 4A). The 1-year survival rate and 3-year survival rate are 84.1% and 27.2%, respectively, in the HBsAg-positive liver graft groups, compared with 95.4% and 79.5%, respectively, in the HBsAg+ liver graft groups, and the differences were statistically significant ( $P = 0.0039$; Fig. 4B). In multivariate analysis, donor HBsAg-positivity was an independent risk factor for survival and incidence ($P = 0.005$ and 0.025, respectively, Table 3).

## DISCUSSION

Organ shortages and waiting list mortality have highlighted the importance of expanding the pool of available livers to save more lives. ECD organs were used in LT including donor with positive serologies for HBV (*Resch et al., 2020*). Previous studies had demonstrated the safety of the utilization of HBsAg positive donors in LT. However, studies on the utilization of HBsAg-positive liver grafts in patients with HCC and posttransplant tumor recurrence are lacking and the safety remains unclear. In this retrospective study, we aim to evaluate the safety and long-term outcome of utilization of HBsAg+ liver grafts using PSM analysis.

The infection of HBV has long been regarded as a major risk factor for the progression of HCC, causing almost 50% cases of HCC in the world (*Jia et al., 2020*). China has the

world's largest burden of HBV infection and therefore HCC is regarded as one of leading causes of tumor-related death in China (*Chen et al., 2016*; *Jing, Liu & Liu, 2020*). In our study, the HBV infection rate of HCC patients reached 100%, and the HBV infection rate of donors was 14.2%. Vaccines and nucleoside drugs have been applied to reduce both the rate of new HBV infection and the progression of liver disease in HBV-positive individuals, which made it possible to use HBsAg-positive donors (*Fanning et al., 2019*). However, its use is controversial, and the recovery of liver function and the risk of relapse are the main concerns after transplantation. Our result showed that the maximum values of ALT, AST and TBIL within 7 days were not significantly higher in the HBsAg-positive liver transplant group, and there was no difference in major early-stage complications and EAD between the two groups. In addition, the incidence of HBsAg-positive was 3.4% overall and 4.2% in the HBsAg-positive transplant groups. From this result, we concluded that with sufficient antiviral prophylaxis and treatment, utilization of HBsAg positive liver grafts in HCC patients had similar liver function recovery and did not increase the incidence of early-stage complications and it was consistent with previous studies on HBsAg-positive donors for LT (*Ballarin et al., 2017*; *Yu et al., 2014*).

Recurrence is the main obstacle limiting posttransplant survival in HCC patients and even with strict selection criteria, the recurrence rate reached up to 20% within 2 years (*Sarici, Isik & Yilmaz, 2020*). Utilization of HBsAg-positive Donors in HCC patients is controversial for HBV infection is an major cause of HCC and is associated with poor prognosis (*Hussain et al., 2017*). *Saab et al. (2009)* concluded that HCC recurrence after LT were related to HBV reinfection and with poorer patient survival. In our study, the result revealed that patient with HBsAg-positive graft had poorer progression-free and overall survival and the HBsAg-positive graft was the independent risk factor. Therefore, the use of HBsAg-positive graft in patients with HCC should be more cautious. This kind of graft may be used in patients with poor condition for salvage transplantation (*Lim et al., 2017*; *Meirelles Junior et al., 2015*). In addition, effective pretransplant treatment and posttransplant monitoring should be applied to reduce recurrence. Early recurrence in HCC patients is typically resulted from micrometastases, which cannot be detected in early stage (*Pantel & Alix-Panabières, 2017*). Our previous study suggested that pretransplant test for circulating tumor cell (CTC) may predict the recurrence of HCC after transplantation (*Chen et al., 2020*). *Wang et al. (2020)* conducted that adjuvant transarterial chemoembolization (TACE) could reduce recurrence of patients with pretransplant positive CTC and this may be applied in LT.

Our study has its limitations. First, the study is a retrospective study from a single center. Larger data and multicenter experience are needed for more convincing results. Second, even PSM analyses were used to reduced selection bias, unidentified confounders still have an influence on the results. Third, sample size is small and a limitation.

## CONCLUSION

In conclusion, this study demonstrated that with sufficient antiviral prophylaxis and treatment, utilization of HBsAg positive liver grafts in HCC patients had similar liver function recovery and did not add the occurrence of early-stage complications. However, patients with an HBsAg-positive graft had poorer progression-free and overall survival and the HBsAg-positive graft was the independent risk factor. Strict selection should be taken for the utilization of HBsAg-positive graft for patients with HCC.

### Funding

This work was funded by the National Natural Science Foundation of China (grand number: 81401324 and 81770410), the Science and Technology Planning Project of Guangdong Province (grand number: 2016A020215048), the Guangdong Provincial International Cooperation Base of Science and Technology (Organ Transplantation) (grant number: 2015B050501002), the Guangdong Provincial Key Laboratory of Organ Donation and Transplant Immunology (grant number: 2013A061401007) and the Guangdong Basic and Applied Basic Research Foundation (grant number: 2020A1515011557, 2020A1515010903). The funders had no role in study design, data collection and analysis, decision to publish, or preparation of the manuscript.

### Grant Disclosures

The following grant information was disclosed by the authors:
National Natural Science Foundation of China: 81401324, 81401324.
Science and Technology Planning Project of Guangdong Province: 2016A020215048.
Guangdong Provincial International Cooperation Base of Science and Technology (Organ Transplantation): 2015B050501002.
Guangdong Provincial International Cooperation Base of Science and Technology (Organ Transplantation): 2015B050501002.
Guangdong Provincial Key Laboratory of Organ Donation and Transplant Immunology: 2013A061401007.
Guangdong Basic and Applied Basic Research Foundation: 2020A1515011557, 2020A1515010903.

### Competing Interests

The authors declare there are no competing interests.

### Author Contributions

- Zhitao Chen conceived and designed the experiments, prepared figures and/or tables, and approved the final draft.
- Yihao Ma performed the experiments, authored or reviewed drafts of the article, and approved the final draft.
- Yuqi Dong analyzed the data, prepared figures and/or tables, and approved the final draft.

- Chuanbao Chen analyzed the data, authored or reviewed drafts of the article, and approved the final draft.
- Hanyu Wang analyzed the data, prepared figures and/or tables, and approved the final draft.
- Tielong Wang performed the experiments, prepared figures and/or tables, and approved the final draft.
- Jia Yu performed the experiments, prepared figures and/or tables, and approved the final draft.
- Xitao Hong performed the experiments, prepared figures and/or tables, and approved the final draft.
- Maogen Chen conceived and designed the experiments, prepared figures and/or tables, and approved the final draft.
- Xiaoshun He conceived and designed the experiments, authored or reviewed drafts of the article, administrative supports, and approved the final draft.
- Weiqiang Ju conceived and designed the experiments, authored or reviewed drafts of the article, administrative supports, and approved the final draft.

### Human Ethics

The following information was supplied relating to ethical approvals (*i.e.*, approving body and any reference numbers):

The study was approved by the Ethics Committee of the First Affiliated Hospital of Sun Yat-sen University (Ethical Application Ref: [2021]437).

### Data Availability

The raw data is available in the Supplemental Files.

### Supplemental Information

Supplemental information for this article can be found online at http://dx.doi.org/10.7717/peerj.15620#supplemental-information.

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
