# Peer review of "Utilization of hepatitis B surface antigen-positive donors in liver transplantation for recipients with hepatocellular carcinoma: a retrospective and propensity score matching analysis"

_PeerJ, doi:10.7717/peerj.15620_

## Round 0.1 · original submission · Minor Revisions

Please address the concerns of both reviewers and amend your manuscript accordingly.

·

Basic reporting

no comment

Experimental design

no comment

Validity of the findings

In this paper, the authors proposed a solution to expand the donor organ source in high prevalence areas and evaluated the safety of utilizing HBsAg positive liver grafts in HCC patients by PSM analysis. This article provides valuable clinical data that is worthy of publication. There are, however, some questions that need to be addressed before it can be seriously considered for publication. Therefore, my temporary decision is minor revision. The detailed comments follow.
(1) It is a good point that the author minimized selection bias and possible cofounders by using PSM analysis. 44 patient pairs out of 338 cases were enrolled and analyzed. May the author explain how these 44 patient pairs and single center data are sufficient to draw the conclusion?
(2) The next most important item is independent risk factors such as HCV, HDV coinfection and Milan criteria. HBV and HCV coinfection have an increased risk for cirrhosis, hepatocellular carcinoma (HCC) (Potthoff, Manns et al. 2010). In addition, HDV is identified as the independent risk factors affecting post-transplant HBV-HCC co-recurrence (Baskiran, Akbulut et al. 2020). I suggest the author provide more information about serological state of the recipient/donor to justify it.
(3) The direct oncogenic effect of the HBV infection may be different between mutated and unmutated HBsAg (Schemmer, Burra et al. 2022). I suggest the authors to consider correlation between HBsAg mutations and progression-free and overall survival. It would be helpful to have some data on antigen mutations so that the relevance can be discussed.
(4) The next point is that the serum HBV load was not released in all the recipients during a follow-up. In the line 164-167” Serovirological test 147 including HBsAg and HBVDNA was monitored at day 7, 14, 30 and then monthly” this would provide valuable data about the safety and serve as a good indicator to HBV recurrence. I suggest the author release these data and make an analysis.
(5) Was there any comparison of outcomes between HBsAg-positive graft recipients with and without HBIg prophylaxis, if appliable?

References:
Baskiran, A., S. Akbulut, T. T. Sahin, C. Koc, S. Karakas, V. Ince, C. Yurdaydin and S. Yilmaz (2020). "Effect of HBV-HDV co-infection on HBV-HCC co-recurrence in patients undergoing living donor liver transplantation." Hepatol Int 14(5): 869-880.
Potthoff, A., M. P. Manns and H. Wedemeyer (2010). "Treatment of HBV/HCV coinfection." Expert Opin Pharmacother 11(6): 919-928.
Schemmer, P., P. Burra, R. H. Hu, C. M. Huber, C. Loinaz, K. Machida, A. Vogel and D. Samuel (2022). "State of the art treatment of hepatitis B virus hepatocellular carcinoma and the role of hepatitis B surface antigen post-liver transplantation and resection." Liver Int 42(2): 288-298.

Additional comments

Miscellaneous corrections:
1. Figure 2, the legend of “59” and “38” was confusing.
2. Please mark the significant level in Figure 2. In the line 183-187, the author did an ANOVA on Fig2, but the significant level is missing.

Reviewer 2 ·

Basic reporting

Overall, the authors have written the manuscript in a clear and well-articulated manner. There were minor grammatical errors at several places. For example, in sentences in line 78, 90 in Introduction, line 184, 190, 198 of Results section, line 224 of Discussion section.

Experimental design

The authors have addressed an interesting research question and used propensity score matching to compare the outcomes of liver transplant in hepatocellular carcinoma patients after receiving grafts from either Hep B surface antigen-positive or -negative donners. The authors conducted extensive analysis using several data points such as Child-Pugh score, tumor grade, tumor recurrence etc. The authors also analyzed postoperative liver function recovery and survival (overall and progression free) between the two groups. The study design is laid out with sufficient details in the flow chart. The methods of analysis are described in sufficient details as well.

Validity of the findings

The authors have described the results and conclusions clearly and concisely. The limitations of the study is also noted at the end. Both the pros and cons of using HBsAg-positive grafts for HCC patients have been discussed.

---

## Round 0.2 · accepted · Accept

All issues pointed by the reviewers were adequately addressed and the manuscript was amended accordingly. Therefore, the revised version is acceptable now.